# Projected Effects of Climate Change on Species Range of *Pantala flavescens*, a Wandering Glider Dragonfly

**DOI:** 10.3390/biology12020226

**Published:** 2023-01-31

**Authors:** Jian Liao, Zhenqi Wu, Haojie Wang, Shaojun Xiao, Ping Mo, Xuefan Cui

**Affiliations:** 1Fisheries College, Guangdong Ocean University, Zhanjiang 524025, China; 2Department of Ecology and Institute of Hydrobiology, Jinan University, Guangzhou 510632, China; 3Guangdong Lianshan Bijiashan provincial Nature Reserve Administration Bureau, Qingyuan 513200, China; 4College of Life and Environmental Sciences, Hunan University of Arts and Science, Changde 415000, China; 5Institute of Hydrobiology, Chinese Academy of Sciences, Wuhan 430072, China

**Keywords:** climate change, global warming, the last glacial maximum, species range, refugia, dragonflies

## Abstract

**Simple Summary:**

In this study, we simulated the distribution range and its shift of *Pantala flavescens* in past, present, and future scenarios, and revealed its habitat properties. Except at high latitudes near the poles (e.g., Antarctica and near the Arctic Circle), it is found almost everywhere in the world, with the most suitable habitat mainly in East Asia and the United States. The max temperature of the warmest month and the precipitation of the wettest month are important factors affecting its distribution, and its suitability decreases with the increase of altitude. Climate warming promoted the shift of lowly and moderately suitable habitats into moderately and highly suitable habitats, especially in equatorial regions, which increased the total habitat area. This study provides a global dynamic distribution pattern of *P. flavescens* across large temporal and spatial scales, and provides a reference for further understanding of its biodiversity and conservation.

**Abstract:**

Dragonflies are sensitive to climate change due to their special habitat in aquatic and terrestrial environments, especially *Pantala flavescens*, which have extraordinary migratory abilities in response to climate change on spatio-temporal scales. At present, there are major gaps in the documentation of insects and the effects of climatic changes on the habitat and species it supports. In this study, we model the global distribution of a wandering glider dragonfly, *P. flavescens*, and detected the important environmental factors shaping its range, as well as habitat shifts under historical and future warming scenarios. The results showed a global map of species ranges of *P. flavescens* currently, including southern North America, most of South America, south-central Africa, most of Europe, South, East and Southeast Asia, and northern Oceania, in total, ca. 6581.667 × 10^4^ km^2^. BIO5 (the max temperature of warmest month) and BIO13 (the precipitation of wettest month) greatly explained its species ranges. The historic refugia were identified around the Great Lakes in the north-central United States. Future warming will increase the total area of suitable habitat and shift the type of suitable habitat compared to the current distribution. The habitat suitability of *P. flavescens* decreased with elevation, global warming forced it to expand to higher elevations, and the habitat suitability of *P. flavescens* around the equator increased with global warming. Overall, our study provides a global dynamic pattern of suitable habitats for *P. flavescens* from the perspective of climate change, and provides a useful reference for biodiversity research and biological conservation.

## 1. Introduction

Climate change and global warming affect biodiversity worldwide and are currently one of the most concerned issues [1]. Fluctuations in climate parameters such as increasing temperatures, rising atmospheric carbon dioxide levels, changing precipitation patterns, and increasing frequency and intensity of extreme climatic events have significant impacts on the distribution of animals and plants [1,2,3]. Future climate warming is indisputable and will have far-reaching and long-term implications, which are becoming more and more visible. In addition, the large-scale shifts of distributions were recognized to take place in response to the development of Quaternary continental ice sheets in history [4]. During the last glacial maximum of the Quaternary, about 26,000 to 19,000 calendar yr ago [5], the cold climate cycles restructured the ecosystem, altered species abundances, and forced many organisms to migrate on a large scale, modifying the distribution patterns of biodiversity [6,7].

Global warming is expected to have a major impact on the pattern of species distribution, a key issue for biological conservation [8,9]. Many species are revealed to shift their ranges in latitude or altitude in response to climate warming [10,11]. A global-scale study revealed a significant trend of more than 1700 species migrating toward the poles, averaging 6.1 km/decade, and significant mean advancement of spring events by 2.3 days per decade [2]. Recently, with global warming, the rate of species migration has increased 2–3 times compared to the past, with the average rate of species migration to higher altitudes being 11.0 m/decade and to higher latitudes being 16.9 km/decade [10]. There was increasing evidence that the distribution of many vertebrates and invertebrates, such as butterflies, dragonflies and damselflies, grasshoppers, lacewings, spiders, herptiles, woodlice, ground beetles, longhorn beetles, soldier beetles, harvestmen, millipedes, aquatic bugs, freshwater fish, birds, and mammals shifted northward and uphill during recent warming periods [2,11,12,13,14,15]. For Odonata insects, a published study presented evidence that 37 species of dragonflies and damselflies from British shifted northwards at their range margins over the past few decades due to climate change [11]. However, little is known about the potential effects of future and more distant historical climates on dragonfly insects.

In the context of global warming, there is a pressing need to understand the ecological properties of species’ habitats and model the distribution status of species in historical periods and predict how the geographical range will change [16,17]. Species distribution models (SDMs) are a useful method for an ecologist to address fundamental questions such as the effects of climate change on the geographic distribution of species [18,19]. The maximum entropy model (MaxEnt) is one of the most recognized and widely used SDMs, which is characterized by rapid operation, high prediction accuracy, and automatic assessment of important environmental factors [16,20,21,22,23]. The MaxEnt model provides an efficient way to model and discriminate between suitable and unsuitable areas based on occurrences and environmental parameters of target species [22]. It has been applied to many aspects, such as biodiversity conservation, medicinal or economic plant development, invasive species management, and genetic geography [24,25,26,27,28,29,30,31].

Dragonflies are widely used in environmental assessments, not only to indicate pollution but also to sense climate change. Generally, species with high-flying abilities have periodic migration behaviors that allow them to spread over long distances. They are also showing positive responses to climate change. *Pantala flavescens* is a common and widespread migratory species of dragonflies, and it usually migrates over long distances with the seasons. However, global climate change is altering its distribution on large temporal and spatial scales. The aims of this study were to investigate the potential global habitats of *P. flavescens*, including historical refugia locations, current distribution of suitable habitats, and future shifts under different warming scenarios, and to explore the important environmental factors shaping the species range.

## 2. Materials and Methods

### 2.1. Study Species

*P. flavescens* (Figure 1) is a wandering glider or global predator known for its extraordinary migrations [32,33]. Its distribution range covers all continents except Antarctica. It can also be found on many islands, but on some remote ones, such as Amsterdam Island in the middle of the Indian Ocean and Easter Island in the Pacific Ocean, it has apparently become a non-immigrant [34,35,36]. The nymphs of *P. flavescens* develop relatively quickly, taking about 40 to 60 days depending on the water temperature [37,38]. Generally, the optimum temperature for the growth and development of *P. flavescens* larvae was above 35 °C [32].

### 2.2. Occurrence Data Collection

Data on locations of the targeted species *P. flavescens*, was obtained by three methods: collected from published literature (see Appendix A), downloaded from the GBIF database (Global Biodiversity Information Facility, GBIF, https://www.gbif.org/ accessed on 7 June 2022), and recorded in our fieldwork from 2011 to 2012 and from 2016 to 2022. For the occurrence records lacking latitude and longitude, we extracted them using Google Earth (https://www.google.com/maps accessed on June 7, 2022) based on the described geographic location. After removing invalid, duplicate, and non-natural records, a total of 5559 records of *P. flavescens* scattered distribution records were obtained. A visual scatter distributional map was developed by ArcGIS 10.7 (Esri, Redlands, CA, USA). A 10 km × 10 km grid was then created, with each grid retaining only one point closest to the center. Finally, 2959 valid records were retained for model analysis (Figure 2).

### 2.3. Environmental Parameters Collection

The environmental parameters included three periods: (1) the Last Glacial Maximum (LGM) represents the historical period; (2) interpolations of observed data represent the current climate; (3) and the simulated 2100 data represent the future climate [39]. For future climate, different shared socio-economic pathways (SSPs, the stable radiation intensities are 2.6 W m^−2^, 4.5 W m^−2^, 7.0 W m^−2^, and 8.5 W m^−2^, respectively) represent four levels of carbon emissions scenarios (CO2-equivalent levels), including the low-end SSP1-2.6 (376 ppm), the middle SSP2-4.5 (650 ppm), the medium-high SSP3-7.0 (1011 ppm) and the worst-case SSP5-8.5 (1228 ppm) [16,40]. Nineteen bioclimatic variables (BIO1 to BIO19) and corresponding elevation data (Ele.) of each period with 2.5 arc-minutes spatial resolution (also known as 5-km spatial resolution) were downloaded from the WORLDCLIM 2.0 (https://www.worldclim.org/data/bioclim.html accessed on June 7, 2022) and used to map the potential habitats of *P. flavescens* and identify the important environmental factors [39].

### 2.4. Statistical Analysis and Suitable Habitat Modeling

In order to reduce autocorrelated bioclimatic factors, we removed the factors (with a VIF value greater than 10) with multi-collinearity by using USDM package version 1.1-18 in R [41]. After filtering, the remaining 11 environmental factors were used for model analysis, including BIO2, BIO3, BIO5, BIO8, BIO9, BIO13, BIO14, BIO15, BIO18, BIO19, and ELE. The Jackknife test was performed to assess the contribution rate (CR) and permutation importance (PI) of environmental factors by using the current data [42]. The filtered environmental factors and occurrence records data were used to predict the potential habitat distribution and range of *P. flavescens* by MaxEnt model version 3.3.3k [22]. Of all the occurrence data used for analysis, 25% was randomly selected for testing and 75% was used for training [16,29,30,43,44,45]. Then set 1000 iterations to run these processes until a threshold of 0.00001 indicates convergence [30,45]. In order to evaluate the performance of MaxEnt model, we first performed a receiver operating characteristic (ROC) analysis and then calculated the value of the area under the receiver operating characteristic curve (AUC). According to the range of AUC values, the performance of the model can be divided into excellent (greater than 0.9), good (between 0.8 and 0.9), fair (between 0.7 and 0.8), poor (between 0.6 and 0.7), and fail (less than 0.6) five levels.

For all the simulated potential distribution areas, we divided it into four different levels of suitable habitats according to the probability of presence (also known as suitability values, ranging from 0 to 1): highly suitable habitat (suitability value greater than 0.6), moderately suitable habitat (between 0.4 and 0.6), and low suitable habitat (between 0.2 and 0.4), as well as unsuitable habitat (less than 0.2) [26,45]. Refugia are established based on the highest historical habitat suitability value (the highest historical value in this study is 0.772), and areas with a historical habitat suitability value greater than 0.75 are considered refugia for this species. The suitable habitat area of each type was calculated by 3D Analyst Tool in ArcToolbox of ArcMap version 10.7 (Esri, Redlands, CA, USA). To test the effects of elevation and latitude on habitat suitability of *P. flavescens*, we used the GAM function of MGCV package version 1.8-40 in R to perform a generalized additive model analysis [46,47].

## 3. Results

### 3.1. MaxEnt Model Performance and Environmental Variable Evaluation

The ROC analysis revealed that the AUC values calculated by training and test data sets were 0.847 and 0.845, respectively, indicating a “good” model performance. The results of the Jackknife test of climatic variables’ contribution are shown in Table 1. According to CR, PI, RTG_O_, TG_O_, and AUC_O_ parameters, max temperature of warmest month (BIO5) was comprehensively evaluated as the most important environmental variable (Table 2). In terms of contribution rate (CR), max temperature of warmest month (BIO5), precipitation of wettest month (BIO13), and mean temperature of driest quarter (BIO9) contributed 83.4%. The permutation importance (PI) identified two important climate variables: max temperature of warmest month (BIO5) and mean temperature of driest quarter (BIO9), which cumulatively accounted for 67.3%. Figure 3 (A and B) were the response curves for the two climate factors that contributed most to the distribution of potential habitats for *P. flavescens*, which were validated with higher training gain and training AUC. When the max temperature of warmest month was between 34 °C and 37 °C, the presence probability of *P. flavescens* was greater than 0.6, indicating a highly suitable habitat, and the suitability value was the highest when the max temperature was 36 °C in the warmest month (Figure 3A). When the precipitation of wettest month range was from 80 to 160 mm, the presence probability of *P. flavescens* was greater than 0.6, and the suitability was the highest when the precipitation was 110 mm.

### 3.2. Current Habitats

The predicted potential species range showed that the highly suitable areas for *P. flavescens* were in various parts of the United States (Gulf Coast, Florida Peninsula, Appalachian Mountains, Lake Huron Basin, Lake Superior Basin, Lake Michigan Basin, Rocky Mountains), eastern China (the North China Plain, the middle-lower Yangtze Plain, Taiwan island, Hainan island, and south-western mountainous area), much of Japan (excepting island of Hokkaido), Vietnam coast, Parana River Basin in South America, the Alps Mountains on the northern shore of the Mediterranean Sea and the Damavand Mountains on the southern shore of the Caspian Sea (Figure 4). Moderate and low suitability habitats were simulated in southern North America, most of South America, south-central Africa, most of Europe, South, East, and Southeast Asia, and northern Oceania (Figure 4). The total potential distribution area covers ca. 6581.667 × 10^4^ km^2^ and contains ca. 388.743 × 10^4^ km^2^ of highly suitable habitats (only 5.90% of the total distribution area) (Table 2).

### 3.3. Historical Habitats and Refugia

Habitat suitability simulation results for a historical period (the Last Glacial Maximum, LGM) are illustrated in Figure 5. The historical habitat distribution was roughly similar to the present, but the area of highly suitable habitats was larger than the current (more highly suitable habitat in East Asia). It was noted that the current South China Sea was the historic Sundaland, which was once a suitable habitat for *P. flavescens* (Figure 5A). The Cordillera and Alaska mountains in North America, Scandinavia in Europe, and the southwest coast of Greenland and the islands in between were also suitable habitats in historical stages (Figure 5A). The area of all potentially suitable habitats in the historical period reached ca. 8029.172 × 10^4^ km^2^, among which the area of high suitability habitat was ca. 752.698 × 10^4^ km^2^ (9.37% of the total area) (Table 2). The refugia of *P. flavescens* was detected in the north-central United States around the Great Lakes (Figure 5B).

### 3.4. Future Habitats in Different Carbon Emission Scenarios

The predicted future suitable habitats of *P. flavescens* under four climate warming scenarios in 2100 were shown in Figure 6. As can be seen from the figure, future warming will increase the area of suitable habitats compared to the current distribution. Potential habitats for expansion include northern Africa, the Arabian Peninsula in western Asia, and southern Oceania, which may be moderate or low suitable habitats for *P. flavescens* in the future. In addition, highly suitable habitats will expand slightly under future warming, and some highly suitable habitats will shift. For example, areas of highly suitable habitats will increase in the Mississippi Plains of the United States (Figure 6), but parts of them will be lost in the Rocky mountains. Moreover, several lowly suitable habitats, such as the East African Plateau and the Congo Basin, will be transformed into moderately suitable habitats in the future. Under SSP1-2.6, SSP2-4.5, SSP3-7.0, and SSP5-5.8 future warming scenarios, the highly suitable habitat area was ca. 406.577 × 10^4^ km^2^, ca. 400.006 × 10^4^ km^2^, ca. 393.321 × 10^4^ km^2^ and ca. 410.056 × 10^4^ km^2^, accounting for 5.03% (ca. 8084.539 × 10^4^ km^2^), 5.06% (ca. 7910.285 × 10^4^ km^2^), 50.6% (ca. 7766.326 × 10^4^ km^2^), and 5.33% (ca. 7699.181 × 10^4^ km^2^) of the total area, respectively (Table 2).

### 3.5. Elevation and Latitude Effects

The elevation effects for historical, current, and four levels of carbon emissions future warming scenarios revealed that the relative suitability of *P. flavescens* habitats decreased with the elevations in both historical and future scenarios (Figure 7A–F). The relative suitability decreased slightly in historical and four future scenarios (Figure 7A,C–F), while it decreased sharply when the elevation exceeds 2700 m a.s.l. in current climate conditions (Figure 7B). In future warming scenarios, the relative suitability of high-elevation habitats will be higher than that of the present. The latitude effect revealed historical and current scenarios in which the suitability of the habitat near the equator was relatively lower than that at mid-latitude (Figure 7G–H). Under four levels of future warming scenarios, there was relatively high suitability for the regions around 30° N, equator and 35° S. However, the suitability of the habitats of *P. flavescens* was relatively low around latitudes 12° N and 18° S, and decreased sharply in areas north of 40° N (Figure 7I–L).

## 4. Discussion

### 4.1. Important Environmental Factors

Climate plays an important role in determining the biological characteristics, properties, and species distribution patterns of plants and animals [48]. Climate is a global phenomenon involving multi-year changes in environmental parameters such as temperature, precipitation, and humidity [1,49]. Temperature is generally considered one of the most critical environmental elements affecting the distribution patterns of many organisms due to the physiological tolerance limits of species [1,2]. The physiological functions of ectotherms can be maintained only within a limited temperature range, nonetheless, when temperatures exceed physiological thresholds, their performance may degrade and even die [50]. As ectotherms, dragonflies are particularly vulnerable to the effects of climate change, such as water temperature fluctuations caused by rising temperatures, which can affect the early larval stages of dragonfly development [3,51]. The larval stage of *P. flavescens* inhabits widely in various shallow-water environments (e.g., ponds, lakes, and rice fields), which are especially susceptible to temperature effects. *P. flavescens* could not survive the winter when the critical water temperature was lower than 14.3 °C [52]. Field and laboratory experiments suggested that temperature warming (15~35 °C) will directly promote the larval growth of *P. flavescens* [32]. However, exceeding its limit temperature will not be conducive to its growth.

Our study found that max temperature of warmest month and precipitation of wettest month well explained the distribution range of *P. flavescens* (69.3% contribution rate, 53.3% permutation importance). Max temperature of warmest month is the upper limit of the temperature of *P. flavescens* in their range, which may seriously affect the physiological function of *P. flavescens* and is not conducive to its survival. It has been reported that the maturation of flight muscles of adult *Libellula pulchella* dragonflies is accompanied by striking changes in thermal physiology, with a mean optimal thoracic temperature (OTT) of 34.7 °C and a mean upper lethal temperature (ULT) of 45.3 °C in newly emerged adults [53]. However, the thoracic temperature of fully mature adults of *L. pulchella* is relatively high, with a mean OTT = 43.5 °C and mean ULT = 48.6 °C [53]. In our study, although we did not examine the thoracic temperature of the flight muscle separately, we found that the max temperature of warmest month (BIO9) in the moderate and high suitable habitat of *P. flavescens* ranged from 23 °C to 44 °C (included the highly suitable habitats ranged from 34 °C to 37 °C), with an optimal BIO9 = 36 °C for suitable habitats. Mean temperature of driest quarter plays another important role in the distribution range of *P. flavescens*. The driest quarter is usually the winter when the temperatures are lower. In temperate regions, populations of adult *P. flavescens* grow rapidly in autumn, but all individuals die in winter for the low air temperature, suggesting low tolerance to low temperatures [52].

Fresh water is essential for dragonflies to obtain resources, grow, and reproduce [16]. Almost all dragonflies lay their eggs in freshwater environments, such as lakes, ponds, rivers, streams, wetlands, or temporary water bodies, and spend their larval stages underwater [16,54]. Although the reproduction and larval development of *P. flavescens* is dependent on freshwater, it seems that *P. flavescens* does not have a high demand for habitat precipitation. In this study, precipitation in wettest month explained 26.3% of the distribution range, and the area with precipitation less than 1200 mm was the moderate or high suitable habitat for *P. flavescens*. The highly suitable habitats have a very narrow precipitation range of 80 to 160 mm in the wettest month. Thus, the most suitable habitats are not those with abundant precipitation. Studies have reported that populations of *P. flavescens* in the Northern Hemisphere typically increase in autumn when there is less precipitation [52]. In short, precipitation, as a crucial climatic factor, affects the distribution range of *P. flavescens*. Precipitation in the wettest month of less than 1200 mm largely explained the distribution of moderate or above suitable habitats for *P. flavescens*, and the precipitation requirement of highly suitable habitat areas is less, ranging from 80 mm to 160 mm.

### 4.2. Suitable Habitat Changes, as Well as Elevation and Latitude Effect

The spatial and temporal dynamics of habitats are crucial for effective conservation of biological resources and have become the focus of global biodiversity and conservation [55,56,57]. Global climate is increasing and shifting the habitat range of many species [16]. *P. flavescens* is an extraordinary migratory dragonfly that lives in tropical and temperate regions and migrates from the tropics to the temperate regions each spring [52], even crossing oceans [37]. For example, *P. flavescens* migrates from India across the Indian Ocean to East Africa in the fall, while the next generation returns from East Africa to India the following spring [37]. These suggest that dragonflies are able to choose favorable winds for migration, but if wind system patterns are affected by climate change, the proposed flight path may be disrupted. Dragonflies are sensitive to climate change, and *P. flavescens* respond to global warming with their remarkable migratory ability, mainly in pursuit of suitable habitats. We summarized the changes in three aspects: habitat area, location, and suitability.

In our study, the habitat area of *P. flavescens* in the historical period (LGM) was ca. 8029.172 × 10^4^ km^2^, including land areas that had disappeared now underwater. For example, Sundaland, once covered with tropical rainforests, was a suitable habitat for *P. flavescens*, which sank below the surface of the South China Sea after the last glacial maximum [16,58,59]. Historically, *P. flavescens* have ranged as far north as the Alaska Mountains in North America, Scandinavia in Europe, and even the southwest coast of Greenland. These areas are close to the Arctic geosphere, and the current climate is too cold for *P. flavescens* to live and breed. Currently, a small number of lowly suitable habitats have been simulated in the UK. Although these are based on the occurrence data of the GBIF database, empiricists believe that *P. flavescens* is still rare in the UK, and even hard to see. Further fieldwork is needed to confirm this. Our projections for four future carbon level warming scenarios suggested that the UK was likely to be invaded, with an increase in lowly suitable habitats and even a few moderately suitable habitats. Additionally, the area of highly suitable habitats during the last glacial maximum was near twice that of the present, mainly because many habitats in East Asia were replaced by moderate habitats under the current climate. In the warming future, four carbon emission scenarios are conducive to the habitat expansion of *P. flavescens*, which is mainly characterized by the increase of moderately suitable habitats and a small amount of lowly suitable habitats (i.e., northern Africa, the Arabian Peninsula in Western Asia, and southern Oceania). In North Africa, moderate and highly suitable habitats are expected to increase in the future, with moderate-low levels of carbon emission scenario SSP2-4.5 showing the largest increase in moderately suitable habitats by 2100, mainly in the Atlas Mountains and the Mediterranean coast. Predicted suitable habitats in the Arabian Peninsula of Western Asia are mainly located near the Suez Canal, Euphrates, and Tigris rivers. Large numbers of *P. flavescens* in the southeastern Arabian Peninsula in October and November may be associated with migrations from more northerly territories, including Middle Asia regions [60,61,62]. *P. flavescens* have also been reported to fly from Middle Asia to East Africa and the Arabian Peninsula [60]. In Oceania, climate warming is expected to expand the population of *P. flavescens* into the southern Wilaber Plain and the Lake Eyre Basin, which seems easy to achieve due to its remarkable migratory ability. However, desert amplification will accelerate over the arid and semiarid regions in the context of severe global warming (e.g., Sahara Desert, Arabian Peninsula, and Great Victoria Desert of Oceania) [63]. Our study supports the view that under the low-end SSP1-2.6 scenario and the low-middle SSP2-4.5 carbon emission scenarios, the area of suitable habitats in these regions increases. However, when greenhouse gas emissions increase to SSP4-7.0 and SSP5-8.5 levels, desertification is aggravated due to the decrease in the vertical exchange of heat, moisture, momentum, trace gases, and aerosols in the surface-atmosphere interface [63]. Thus, the trend of habitat expansion in North Africa and the Arabian Peninsula under the scenarios of SSP4-7.0 and SSP5-8.5 in 2100 will be reduced, and the warming in the Great Victoria Desert and the Great Desert areas in Oceania will strengthen desertification, which is not conducive to the habitat of *P. flavescens*.

Upward and poleward shifts are the most common types of species range shifts reported in response to climate change [14]. In our study, we found that *P. flavescens* tends to expand to higher elevations under future warming scenarios. But unlike most other published species, future warming leads to an increase in its habitat suitability near the equator, indicating a possible expansion toward the equatorial region. Under the current climate conditions, the relative suitability of habitat decreased sharply above 2700 m, indicating that it would not be conducive for *P. flavescens* to survive here. Environmental thermal conditions affect the larval growth of *P. flavescens*, and the low temperature at high elevations is the key factor limiting larval development [32]. However, the trend of habitat suitability decreasing with elevation will be alleviated by future warming scenarios, viz., *P. flavescens* will expand to higher elevations under a warming future. The warming climate is likely to have directional effects on species range shifts because temperatures are spatially autocorrelated, linking warmer conditions at lower elevations or latitudes with cooler conditions at higher elevation latitudes [14]. Thus, upward dispersal becomes possible. However, poleward dispersal is not as expected. Instead, habitats near the equator will be increasingly suitable for *P. flavescens* under warming scenarios. For example, the equatorial East African Plateau and Congo Basin are mostly low suitable habitats for *P. flavescens* under current climate scenarios, which will be transformed into moderately suitable habitats with global warming. The East African Plateau and the Congo Basin are both near the equator, but the Congo Basin is in the west of the continent, with low terrain. Currently, it is affected by the equatorial low atmospheric pressure and southwest wind all the year-round, with high temperature and rain. Excessive rainfall is not conducive to the emergence of *P. flavescens*, so it also inhibits its maturation and reproduction. However, the East African Plateau (1000~1500 m a.s.l.) has relatively higher terrain and less rainfall, which is more suitable for the survival of *P. flavescens*. The East African Plateau has the highest concentration of lakes in Africa, including Lake Victoria, the largest lake in Africa, and other lakes of various sizes [64], providing heat and moisture balance for *P. flavescens*. In the warming future, rising temperature will promote the habitat autocorrelation between the East African Plateau and the Congo Basin, which will transform lowly suitable habitats into moderately suitable habitats.

### 4.3. Historical Refugia

Refugia are important in thinking about shifts in species distribution, particularly in response to historical climatic changes, such as the Last Glacial Maximum [4]. Refugia are habitats where biodiversity retreat persists, and may expand from there under changing climatic conditions [65], facilitating the survival of organisms during periods of extreme climatic change [66,67]. Our study identified refugia of *P. flavescens* during the Last Glacial Maximum near the Great Lakes in the north-central United States. This was once the last habitat of *P. flavescens* and the initial habitat for post-glacial species redistribution, from which species migrated. Located in North America, the Great Lakes are the largest lakes on earth in terms of surface area and freshwater reserves [68]. Not only are the Great Lakes sensitive to climate change, which may have contributed to large recent fluctuations in lake water level, thermal structure, and ice coverage, but they are also an important regional climate driver due to their large size, thermal inertia, and surface area [69,70,71,72]. The Lower Great Lakes, Lake Erie, and Ontario, have been studied as a refuge for the Zebra mussel [73]. The Great Lakes are also a refuge for many species of fish, such as lake trout, ciscoes, walleyes, and a few non-managed species [74]. In addition, studies have shown that rivers in the northeastern United States (around the Great Lakes) can serve as climate refuges for riverine dragonflies, but future warming is expected to reduce the available habitat for these riverine dragonflies [75].

Current species ranges are greatly influenced by the location of historical refugia and the dispersal process [4]. This study revealed the refugia location of *P. flavescens* in the historical stage, which helped us to understand the distribution characteristics of *P. flavescens* to a great extent. Dispersal is an important ecological process that not only promotes gene flow among populations, but also reshapes the distribution pattern of species at the genetic level. Dispersal ability varies from species to species, and so does the pattern of distribution shaped by it. Published research suggested that a weak flying damselfly, *Neurobasis chinensis* retreated to south-central Vietnam after shrinking its habitat during the LGM period, and gradually spread from south-central Vietnam to Southeast Asia, South Asia and East Asia over millions of years, forming its current distribution pattern [16]. In addition, most species of the genus *Neurobasis* are so weak-flight that they are now confined to a few Southeast Asian islands, such as *Neurobasis kaupi* live only on the island of Sulawesi, *Neurobasis florida* on Java of Indonesia, and *Neurobasis daviesi* on the island of Palawan in the Philippines [16,76]. Although climate change has tried to alter their distribution pattern, the genus of *Neurobasis* is limited by its ability to spread and its habitat’s dependence on streams. For our research objects, *P. flavescens* is a good flier, and has low requirements in habitats. It mainly distributes in some shallow water habitats and even temporary water bodies. Its strong dispersal ability and adaptability lead to its susceptibility to climate change.

## 5. Conclusions

The first suitable habitat distribution map of *P. flavescens* was simulated by MaxEnt model under the current climate environment, including southern North America, most of South America, south-central Africa, most of Europe, South, East and Southeast Asia, and northern Oceania, in total, ca. 6581.667 × 10^4^ km^2^. Highly suitable habitats were detected in various parts of the United States (Gulf Coast, Florida Peninsula, Appalachian Mountains, Lake Huron Basin, Lake Superior Basin, Lake Michigan Basin, Rocky Mountains), eastern China (the North China Plain, the Middle-lower Yangtze Plain, Taiwan island, Hainan island and southwestern mountainous area), much of Japan (excepting island of Hokkaido), Vietnam coast, Parana River Basin in South America, the Alps Mountains on the northern shore of the Mediterranean Sea and the Damavand Mountains on the southern shore of the Caspian Sea. We also detected that the historical refuge of *P. flavescens* was located around the Great Lakes in the north-central United States. Future warming will increase the total area of suitable habitats compared to the current distribution, and shift the type of suitable habitats. The max temperature of warmest month (BIO5) and the precipitation of wettest month (BIO13) are the two most important climatic factors that shape and shift the habitat distribution of *P. flavescens*. The habitat suitability of *P. flavescens* decreased with elevation, global warming forced it to expand to higher elevations, and the habitat suitability of *P. flavescens* around the equator also increased with global warming. The projected spatial and temporal pattern of range shifts and the refugia location of *P. flavescens* will be useful references in developing environmental management and biological conservation strategies.

## Figures and Tables

**Figure 1 biology-12-00226-f001:**
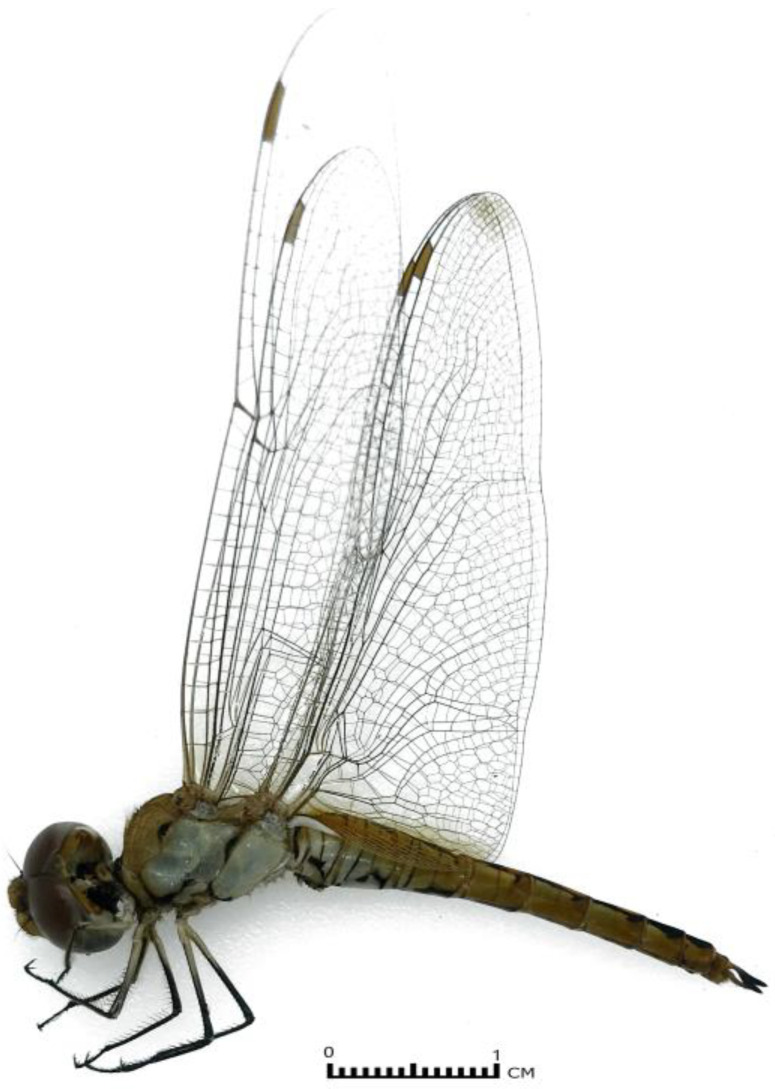
Specimen of *P. flavescens* (male) was collected from Nankunshan Provincial Nature Reserve, Guangdong, China in June 2021.

**Figure 2 biology-12-00226-f002:**
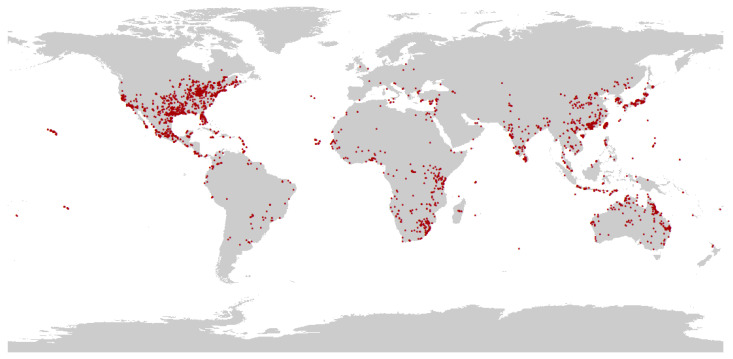
Occurrence recorded of *P. flavescens* in the world.

**Figure 3 biology-12-00226-f003:**
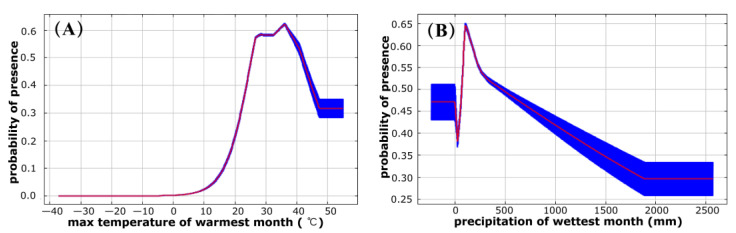
Response curves of presence probability of *P. flavescens* to max temperature of warmest month (**A**) and precipitation of wettest month (**B**).

**Figure 4 biology-12-00226-f004:**
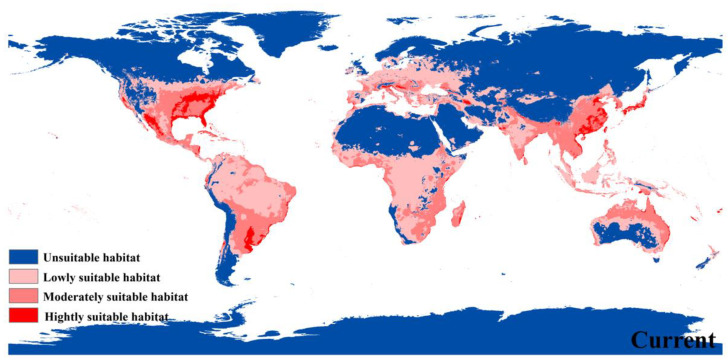
Habitat suitability distribution of *P. flavescens* according to occurrence records.

**Figure 5 biology-12-00226-f005:**
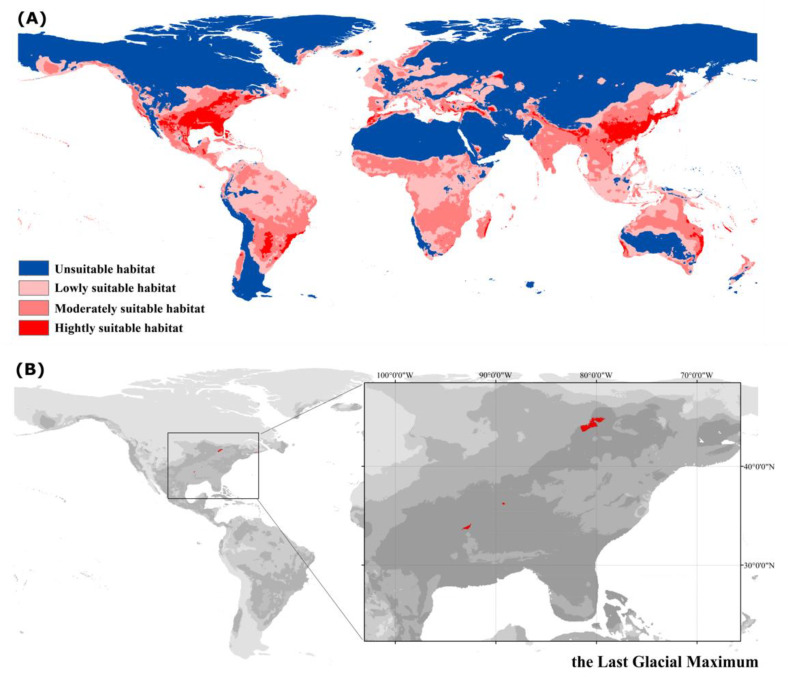
Modeling of suitable habitat distribution (**A**) and refugia (**B**) of *P. flavescens* in historical period: the Last Glacial Maximum. Light gray to dark gray represents an increase in habitat suitability, and red represents the location of refugia.

**Figure 6 biology-12-00226-f006:**
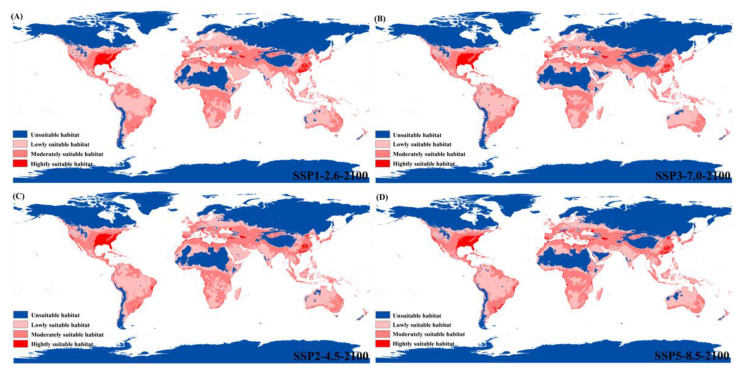
Habitat distribution of *P. flavescens* under four carbon emission levels in 2100. (**A**) SSP1-2.6: the low-end level; (**B**) SSP2-4.5: the low-moderate level; (**C**) SSP3-7.0: the medium-high level; (**D**) SSP5-8.5: the high SSP5-8.5 level.

**Figure 7 biology-12-00226-f007:**
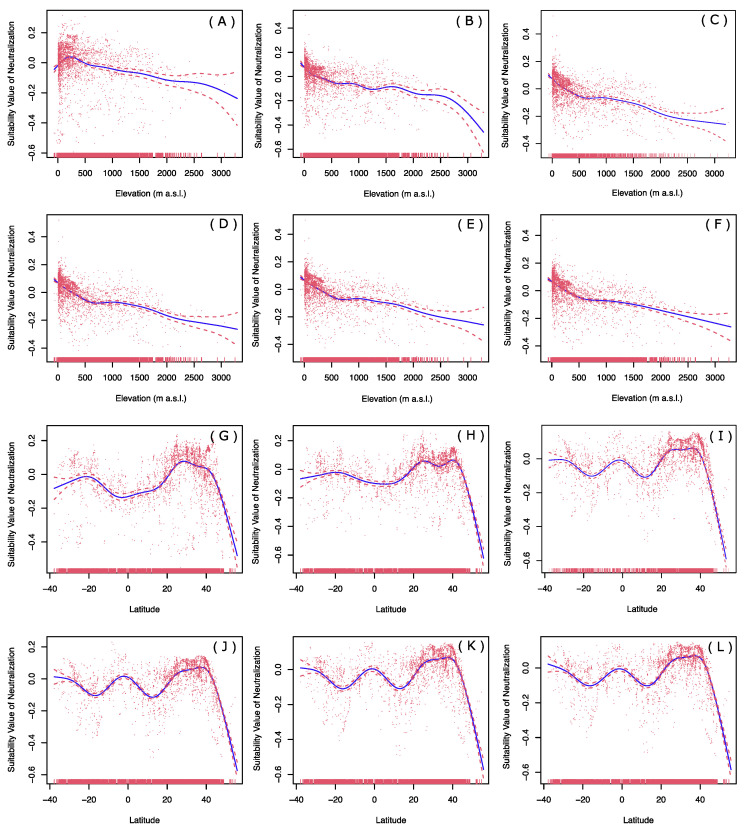
Changes in elevation and latitude of *P. flavescens* distribution over long time scales based on generalized additive model. All suitability values are neutralized by GAM function. (**A**) Elevations vs historical suitability value; (**B**) elevation vs current suitability value; (**C**) elevation vs future SSP1-2.6 suitability value; (**D**) elevation vs future SSP2-4.5 suitability value; (**E**) elevation vs future SSP3-7.0 suitability value; (**F**) elevation vs future SSP5-8.5 suitability value; (**G**) latitude vs historical suitability value; (**H**) latitude vs current suitability value; (**I**) latitude vs future SSP1-2.6 suitability value; (**J**) latitude vs future SSP2-4.5 suitability value; (**K**) latitude vs future SSP3-7.0 suitability value; (**L**) latitude vs future SSP5-8.5 suitability value.

**Table 1 biology-12-00226-t001:** The contribution rate (CR), permutation importance (PI) of filtered environmental factors used in the model prediction, and relevant information of Jackknife test. RTG_W_, regularization training gain without this variable; RTG_O_, regularization training gain only with this variable; TG_W_, test gain without this variable; TG_O_, test gain only with this variable; AUC_W_, AUC without this variable; AUC_O_, AUC only with this variable.

Variables	CR (%)	PI (%)	RTG_W_	RTG_O_	TG_W_	TG_O_	AUC_W_	AUC_O_
BIO2	0.8077	2.9101	0.8348	0.0639	0.8770	0.0711	0.8426	0.6073
BIO3	4.1241	5.0757	0.8318	0.3587	0.8778	0.3684	0.8429	0.7075
BIO5	42.9567	48.1969	0.8115	0.6087	0.8539	0.6304	0.8393	0.7781
BIO8	0.6035	2.3714	0.8432	0.4787	0.8905	0.4998	0.8448	0.7606
BIO9	14.1083	19.1390	0.8327	0.5518	0.8784	0.5691	0.8438	0.7566
BIO13	26.3270	5.1150	0.8326	0.4763	0.8808	0.4852	0.8434	0.7447
BIO14	2.4172	3.4321	0.8409	0.2987	0.8846	0.3135	0.8438	0.7146
BIO15	0.4081	5.6594	0.8359	0.1045	0.8756	0.1151	0.8418	0.6125
BIO18	3.6003	1.3327	0.8407	0.4221	0.8842	0.4356	0.8439	0.7490
BIO19	2.7809	4.6792	0.8357	0.2384	0.8801	0.2505	0.8431	0.6892
ELE	1.8663	2.0886	0.8326	0.2400	0.8746	0.2486	0.8417	0.6783

**Table 2 biology-12-00226-t002:** Change of suitable habitat (×10^4^ km^2^) of *P. flavescens* in historical, current, and four levels of future scenarios.

Scenario	Highly Suitable Habitat	Moderately Suitable Habitat	Lowly Suitable Habitat	Total
**Historical**	752.698	3528.225	3748.250	8029.172
**Current**	388.743	2453.779	3739.145	6581.667
**SSP1-2.6**	406.577	3651.409	4026.553	8084.539
**SSP2-4.5**	400.006	3634.743	3875.535	7910.285
**SSP3-7.0**	393.321	3648.141	3724.864	7766.326
**SSP5-8.5**	410.056	3636.386	3652.739	7699.181

## Data Availability

The data presented in this study are available on request from the corresponding author.

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
