# Peer review of "Projected Effects of Climate Change on Species Range of Pantala flavescens, a Wandering Glider Dragonfly"

_biology, 2023, doi:10.3390/biology12020226_

Round 1

Reviewer 1 Report

Line 173 and 175: write only 1 decimal (83.4% and 67.3%).

Line 225: P. flavescens in cursive/italic.

Line 287: Wrong citation. Change (Augustine et al., 2021) to [32].

Line 290 and 291: write only 1 decimal (69.3% and 53.3%).

Line 309: Wrong citation. Change (Liao et al., 2022) to [16].

Line 314: write only 1 decimal (26.3%).

Line 315: P. flavescens in cursive/italic.

Line 316: add a space in 160mm.

Line 335: change 8029.172×104 km2 to 8029.172×104 km2

Line 533: Pantala flavescens in cursive/italic.

Line 535: Pantala flavescens in cursive/italic.

Line 541: Pantala flavescens in cursive/italic.

Line 548: Pantala flavescens in cursive/italic.

Line 581: Pantala flavescens in cursive/italic.

Line 598: Pantala flavescens in cursive/italic.

Author Response

Thank you for your comments and suggestions. I have modified the format and citation according to your suggestions.

Reviewer 2 Report

A very interesting paper on a species that has been described as arguably the most successful dragonfly in the world (Dijkstra & Lewington). This is an example of the efficient use of statistical tools to model possible changes in species distribution based on current data. Pantala flavescens is already appearing in many new regions, and in the future it may significantly remodel dragonfly assemblages and local biota. Therefore, the analyses presented in this paper are not just a methodological exercise. They can also be a model of changes in the distribution of other species, which are more difficult to describe due to the smaller amount of data.

The evaluated paper is methodologically correct and technically well prepared. Minor errors are noted in the attached manuscript.

I only have one significant critical remark. The authors wrote: “Data on locations (...) was obtained by three methods: collected from published literature...”.

First, it means all or nothing. All publications from world literature with data on Pantala flavescens? Any number of randomly selected texts? Publications selected from a certain angle? If so, what were the criteria?

Secondly, the authors do not cite anything here. It is unacceptable. It was necessary to list (cite) all the sources of this kind used. There are at least two reasons. The first reason is ethical: the use of published data is obligatorily associated with citing their authors. Otherwise, we break the good practice of science and do not respect the authors of publications as their creators.  The second reason is substantive: without a list of works used, the reviewer (and then the reader) cannot assess whether all relevant publications have been included. There are a lot of publications about Pantala flavescens and it is easy to miss something important even during a careful literature search. It may also happen that unreliable publications are included.

I am curious if the UK records are based on new data. Even in the latest publications there is no data on the occurrence of Pantala flavescens in the wild in the British Isles. Several older records were caused by the accidental introduction of imagines on ships or with loads of bananas, so they are not included in the distribution maps of the species in European odonatological monographs. I know there are four UK records in the GBIF. However, GBIF also includes literature, which can be seen, for example, in data from Central and Central and Eastern Europe. So the question is: is this the legacy data I am writing about? Or are these new observations from the UK that have not yet been published? This is data at the edge of the area of occurrence, which may affect the modelling results.

The discussion could use the fact that the modelling results (predictors of the area of occurrence of Pantala flavescens) agree with the causes of seasonal migrations of this species with monsoons between Africa and Asia, already described in the literature. These data fit together beautifully.

In the last paragraph of the “Introduction”, the authors move a little too abruptly from the general to the specific. It is worth pointing out directly why Pantala flavescens was chosen. I know that this is supposed to result from the previous paragraphs, but one could just as easily analyse, for example, Aeshna affinis, Anax parthenope, Crocothemis erythraea, Sympetrum fonscolombii or Orthetrum albistylum. Pantala flavescens is of course particularly interesting, as I wrote in the first paragraph of this review.

Author Response

Thank you for your detailed comments and suggestions, which are very helpful to our manuscript. We tried to revise the manuscript according to your suggestions. We added the literature and methods of collection. We explained the distribution of UK and added content in the discussion section. About why we choose P. flavescens, we added the connecting sentence in the last paragraph. Detailed responses can be found in the word document. Thanks again!
